# OpenReview forum: "When to Make Exceptions: Exploring Language Models as Accounts of Human Moral Judgment"
_NeurIPS.cc/2022/Conference — NeurIPS 2022 Accept_

### Official Review · Reviewer_bKwH · 2022-07-07

**Rating:** 7
**Confidence:** 3
**Soundness:** 2 fair
**Presentation:** 3 good
**Contribution:** 3 good

**Summary:**

This paper addresses the important question whether LLMs understand human flexible moral judgements. Human moral judgements are based on rules (like "don't cut the line"), but most important are flexible. Depending on the situation, it might be morally OK to break the rule. Prior work (like MoralQA) doesn't address this important aspect of moral judgements.
This work proposes a challenge set that addresses precisely this. The set contains three different rules, each with a set of situations that either allow breaking them or not. The "ground-truth" is a large set of judgements made by humans. The situations are made such that novel scenarios occur that are unlikely to have occurred in training. This is important, because central to human morality is the ability to respond in a novel situation that is hasn't been exposed to in training.
The authors evaluate a set of language models on this challenge set, most notably of which Delphi (domain-specific, trained on moral judgements) and instructGPT (one of the SOTA LLMs). They show that most models don't achieve much better than random on this task, but that their proposed method called moral chain-of-thought prompting improves the SOTA by ~11% F1. The authors do an extensive error analysis and include high-stake wrong predictions in the appendix (e.g. when the model doesn't realise a life being at stake is important).

**Questions:**

Questions that could address limitations:
- How sensitive is the SOTA model (MoralCoT) to prompt wording and ordering?
- Would it be possible to add some conclusions from analyses that are part of the appendix in the main text?
- Same but for human annotator demographic (seems like things could be cultural dependent, i.e. in some cultures it might be absolutely not OK to step on money when it is about to be blown away by the wind, whereas in others that's considered helpful)
- Do you have any data on how much humans agree usually in the judgements, i.e. what is the distribution of percentages of permissibility?

General suggestions:
- Consider also citing the following work for Chain-of-Thought prompting (scratchpad): https://arxiv.org/abs/2112.00114
- s/that/those line 74
- 131-133: I would rephrase this a bit less strong, it seems to me like the previous setup (moralQA + finetuning) can test flexibility (simply by having questions that reflect flexible judgement), but they don't currently, and your data is in a sense complementary; moralQA can test whether the model even has knowledge of the vast amount of norms out there, you test wether they understand the flexibility but with less data/norms.
- typo line 206
- I don't fully understand what's going on in the line 259 within the `concat()` operation
- Reference to table 8 in line 312 should be table 4

**Limitations:**

Yes, the authors talk about the dataset size, the fact that there are different theories of moral judgement, the fact that models shouldn't be making moral decisions automatically.

**Strengths And Weaknesses:**

**Strengths**
- This paper addresses a very significant subject (to what extent do the LLM align with human moral judgements)
- It does so by realising that this is not an objective task. The moral judgements are made by a large group of human annotators and not only binary accuracy by the models is reported (OK to break rule vs not OK), but also the % of people saying it's OK.
- If it's true that a challenge set like this (flexible moral judgements) doesn't exist yet (reviewer can't judge because doesn't know the literature), then it's a clearly important and original addition.
- The clarity of writing is good, the paper is easily understood in a single pass.
- The dataset construction is sound and contains a lot of human judgements.

**Weaknesses**
- Information about the type of human annotators is missing (demographic, how were they recruited, etc.)
- It's quite difficult to judge the results without looking at the appendix. E.g. in "checking subquestion answers" I have no idea what this accuracy means. Mostly I would at least add a conclusions derived from the results in the main paper.
- It's a well-known issue that LLMs are sensitive to prompt wording and even ordering ([1], [2]). Although the authors do mention they did an analysis on the sensitivity on the model to this and add it in the appendix, to make the claims that this paper makes I think this should be part of the main results (i.e. in the form of variance over prompt wording in Table 3).

[1] Ethan Perez, Douwe Kiela, and Kyunghyun Cho. True few-shot learning with language models. In A. Beygelzimer, Y. Dauphin, P. Liang, and J. Wortman Vaughan (eds.), Advances in Neural Information Processing Systems, 2021. URL https://openreview.net/forum?id=ShnM-rRh4T
[2] Yao Lu, Max Bartolo, Alastair Moore, Sebastian Riedel, and Pontus Stenetorp. Fantastically ordered prompts and where to find them: Overcoming few-shot prompt order sensitivity. In ACL, 2022.

---

> ### Author Response · Authors · 2022-08-02
> **Response to Reviewer bKwH**
>
> We thank the reviewer for the valuable comment, pointing out that "this paper addresses a very significant subject", "dataset construction is sound" and "the paper is easily understood".
>
> > It's quite difficult to judge the results without looking at the appendix. Mostly I would at least add a conclusions derived from the results in the main paper.
>
> We appreciate the reviewer's suggestion. Due to the page limit, lots of details went to the appendix in the submitted version of the paper. In camera ready, we will make use of the extra one page to add conclusions derived from the results in the main paper, and also add key implementation details into the main paper.
>
> > It's a well-known issue that LLMs are sensitive to prompt wording and even ordering ([1], [2]). Although the authors do mention they did an analysis on the sensitivity on the model to this and add it in the appendix, to make the claims that this paper makes I think this should be part of the main results (i.e. in the form of variance over prompt wording in Table 3). / How sensitive is the SOTA model (MoralCoT) to prompt wording and ordering?
>
> Thanks for this suggestion! We will move the performance reported in the appendix to the main results in the form of standard deviation in the next version of the paper.
>
> > Information about the type of human annotators is missing (demographic, how were they recruited, etc.) / For human annotator demographic (seems like things could be cultural dependent, i.e. in some cultures it might be absolutely not OK to step on money when it is about to be blown away by the wind, whereas in others that's considered helpful)
>
> Thanks so much for pointing this out.  We completely agree that demographic information of our subjects/annotators is critical information for studying morality because at least some moral judgments may be culturally dependent.  Here is a brief synopsis of the demographic features of our subjects.  We will add this information into the supplementary.
>
> Novel rule study: Participation in the study was limited to MTurk workers located in the US.  Mean age = 37.2 years, SD age = 11.9 years.  Race/ethnicity: 68.5% white, 10.1% asian, 6.0% black, 5.7% Hispanic, Latino or Spanish Origin, 10.7% mixed race or other.  Mean political leaning was 3.4 on a 5-point scale, anchored at 1 (extremely conservative) and 5 (extremely liberal).
>
> Snack study 1: Participation in the study was limited to MTURK workers located in the US.  Mean age =38 years, SD age = 11.0 years.  Race/ethnicity: 80.3% white, 4.2% Asian, 12.7% Black or African American, 7.0% Hispanic, Latino or Spanish Origin, 1.5% other (categories are not exclusive of one another; percents sum to more than 1).  Mean political leaning was 3.1 on a 5-point scale, anchored at 1 (extremely conservative) and 5 (extremely liberal).
>
> Snack study 2:  Participation in the study was limited to MTURK workers located in the US.  Mean age = 37.1 years, SD age = 10.4 years.  Race/ethnicity: 76.9% White, 5.0% Asian, 14.0% Black or African American, 6.6% Hispanic, Latino or Spanish Origin, 5.0% other (categories are not exclusive of one another; percents sum to more than 1).  Mean political leaning was 3.6 on a 5-point scale, anchored at 1 (extremely conservative) and 5 (extremely liberal).
>
> Property damage & deli line subjects: Participation in the study was limited to MTURK workers located in the US.  No further demographic data was taken from participants, but average demographic information for MTURK participants is as follows (Difallah, Filatova, & Ipeirotis, 2018). Gender: 55 % Female. Age: 20% born after 1990, 60% born after 1980, and 80% born after 1970. Median household income: $47K/year.
>
> > Do you have any data on how much humans agree usually in the judgements, i.e. what is the distribution of percentages of permissibility?
>
> The distribution of percentages is as follows:  0-10: 14.86% of the data, 10-20: 12.16% of the data, 20-30: 11.49%, 30-40: 9.46%, 40-50: 12.84%, 50-60: 4.73%, 60-70: 6.76% , 70-80: 9.46%, 80-90: 11.49% , 90-100: 6.76%
>
> ### Writing Suggestions
> > - Consider also citing the following work for Chain-of-Thought prompting (scratchpad): https://arxiv.org/abs/2112.00114
> > - s/that/those line 74
> > - 131-133: I would rephrase this a bit less strong [...]
> > - typo line 206
> > - Reference to table 8 in line 312 should be table 4
>
> Thanks for the careful suggestions. We have incorporated the suggested changes to the revised PDF.
>
> > I don't fully understand what's going on in the line 259 within the concat() operation
>
> The concat() operation means natural language concatenation. The main goal is to elicit the model to answer subquestions and then combine all these answers to form a final query, so this concatenated query is like "[Vignette] [subquestion 1] [answer to subquestion 1] [subquestion 2] [answer to subquestion 2] ... Taking all these into account, is it OK for that person to break the rule in this case?"

---

> > ### Comment · Reviewer_bKwH · 2022-08-05
> > **Thanks!**
> >
> > Thanks for the response and incorporating this important extra information in the revision. My questions have been adequately addressed

---

### Official Review · Reviewer_P8j4 · 2022-07-08

**Rating:** 6
**Confidence:** 4
**Soundness:** 3 good
**Presentation:** 3 good
**Contribution:** 3 good

**Summary:**

This paper addresses the problem of improving language models by taking into account human moral judgment: basically, the idea is to build AI systems that have to establish whether it is permissible to break a moral rule in several situations. This is a crucial task in AI safety, as it deals with the capability of understanding ethics in relation to heterogeneous contexts. The proposed approach consists in a prompting strategy that generates a sequence of questions and answers that can be exploited to predict human moral judgment.


**Questions:**

* How is the sequence of questions q_1, ..., q_N chosen? Is it fixed (i.e., pre-determined) or adapted to each case?

* Did you try other aggregation functions rather than concatenation to obtain the chained prompt c_i?

**Limitations:**

The main limitation of the paper is that the decription of the model is quite short. I would have liked to see more details on the cognivitely-inspired model: for example, whether the posed questions are fixed or not, or whether the authors tested a different aggregation function rather than concatenation (e.g., similarly to what happens with memory networks).

Other comments and typos:
-- In Table 3, the last three columns are misleading: they represent an F1-score, but this becomes clear only after reading the caption. I would suggest either to split the table in two, or to move the three columns next to the initial F1 column, and include a supercolumn named F1 covering all the four cases (the average, plus the three single scenarios).
-- Line 255, "to a multi-step prompt in Figure 1" -> "to a nulti-step prompt, as shown in Figure 1"
-- Line 344, "that contradicts" -> "that contradict"

**Strengths And Weaknesses:**

+ Building AI systems that account for human moral judgment is a crucial need
+ The motivation related to AI safety is particularly strong
+ Rule-Breaking Question Answering (RBQA) is a new challenging task/dataset for NLP systems that aim to incorporate human moral judgment

- The description of the model and of the proposed approach is too brief, and several details are lacking or just mention without a proper analysis.

---

> ### Author Response · Authors · 2022-08-02
> **Response to Reviewer P8j4**
>
> We thank the reviewer for their review, and the generally positive assessment of our work, -- “building AI systems that account for human moral judgment is a crucial need”, “The motivation related to AI safety is particularly strong”, and “RBQA is a new challenging task/dataset”.
>
> ### Technical Details:
> > How is the sequence of questions q_1, ..., q_N chosen? Is it fixed (i.e., pre-determined) or adapted to each case?
>
> In this study, the sequence of questions q_1, ..., q_N are a pre-determined list of questions, because adapting the questions to each case might require training data, which is not the scope for this work. (We encourage this direction of exploration in future work.)
>
> In this work, the pre-determined set of questions (as shown in Figure 1) was chosen based on experiments that have been conducted with human subjects.  The questions were based on factors that moral psychologists have found to be important predictors of the moral judgments that humans make.  The questions from the psychology studies were themselves inspired by different schools of moral philosophy (e.g., the deontology question reflects whether there is a pre-existing rule, the contractualist question considers the purpose of the rule, and the consequentialist question evaluates the benefit and harm of breaking the rule).
>
> > Did you try other aggregation functions rather than concatenation to obtain the chained prompt c_i? / whether the authors tested a different aggregation function rather than concatenation (e.g., similarly to what happens with memory networks)
>
> This work contributes a challenge set, but not a training set. So different aggregation functions as in the memory network can hardly be trained, as the technical realization of them is often to open up the large language model, and train their parameters, or the additional layer that process the aggregated embedding. For example, if the aggregation method is to add up the embeddings of the answers to the subquestions, then the model needs to learn an extra layer to map this aggregated embedding to the classification label.
>
> ### Writing and formatting suggestions:
>
> > Other comments and typos: -- In Table 3, the last three columns are misleading: they represent an F1-score, but this becomes clear only after reading the caption. I would suggest either to split the table in two, or to move the three columns next to the initial F1 column, and include a supercolumn named F1 covering all the four cases (the average, plus the three single scenarios).
>
> Thank you for the suggestion on more clear representation. We have reformatted Table 3 by adding a separator line and the super column names and revised the paper PDF. In the camera ready version, it will look nicer if splitting the table into two. We will do the split using the extra one page in the camera ready.
>
> > -- Line 255, "to a multi-step prompt in Figure 1" -> "to a nulti-step prompt, as shown in Figure 1" -- Line 344, "that contradicts" -> "that contradict"
>
> We have also fixed both typos that the reviewer carefully pointed out.

---

### Official Review · Reviewer_eUQ2 · 2022-07-11

**Rating:** 5
**Confidence:** 3
**Soundness:** 3 good
**Presentation:** 3 good
**Contribution:** 3 good

**Summary:**

To achieve capturing the flexibility of the human moral mind, authors created a challenge set consisting of rule-breaking question answering.
They use the state of the art large language model (InstructGPT) and developed MoralCOT that is based on theories of moral reasoning, which outperformed other existing large language models, they say.


**Questions:**

- I think relying on InstructGPT is problematic for reproducibility, I'm wondering if InstructBERT can be compared?
   I would at least like to see the results of applying the same mechanism as MoralCOT to a pretrain model such as BERT.

- For binary predictions of human moral judgement, how the inter-rater correlation would be? I'm wondering if each model could be comparable to individual's assessment.

**Limitations:**

Authors employed only 3 types of norms, which could be regarded as preliminary study Therefore they need to prove the types are sufficient to cover the human morality. At least they should show how these 3 types of norms are important for dealing with human morality.

**Strengths And Weaknesses:**

[Strength]
In my opinion, authors' ideas (e.g., breaking down into a few decision processes and estimating costs) are quite novel from the psycholinguistic view point. They also provide detailed examples of input and output, which may be beneficial for the community.

[Weaknesses]
Dealing with only 3 types of norms are not very sufficient.
The proposed method fully relies on open AI's API, which leads less reproducibility.
Details of human's responses are not shown, which results in less clarity.

---

> ### Author Response · Authors · 2022-08-02
> **Response to Reviewer eUQ2**
>
> We thank the reviewer for pointing out the novelty of our work “authors' ideas are quite novel from the psycholinguistic view point”, and their generally positive assessment of our work.
>
> > Authors employed only 3 types of norms, which could be regarded as preliminary study. Therefore they need to prove the types are sufficient to cover the human morality. At least they should show how these 3 types of norms are important for dealing with human morality.
>
> This work uses only three norms because our approach is to explore each individual norm thoroughly in order to understand the underlying structure of the way that these norms can be permissibly violated.  We therefore chose a small number of norms but probed dozens of ways that the norm might be violated.
>
> In addition, each rule acts as a case study of the broader category of rules that they represent.  These categories are 1) rules that are found cross culturally (and therefore may be universal to human morality), 2) rules that vary cross culturally and are therefore constructed in social contexts, and 3) rules that are invented to adjudicate completely novel contexts.  These three categories represent rules that need to be reasoned about using three distinct kinds of moral cognition -- that supported by socio-cultural evolution, that supported by social learning, and that supported by individual reasoning alone. So, if a model succeeds on RBQA, it would suggest that the model has achieved an important competence.  We will elaborate on this idea in the final version (and cite another paper of ours in progress which draws out this distinction).
>
> > The proposed method fully relies on open AI's API, which leads to less reproducibility. / I think relying on InstructGPT is problematic for reproducibility, I'm wondering if InstructBERT can be compared? I would at least like to see the results of applying the same mechanism as MoralCOT to a pretrain model such as BERT.
>
> We think maybe the reviewer is asking about "generalizability" rather than "reproducibility". For reproducibility, we release our codes, and use temperature=0 (i.e., argmax when decoding) so that InstructGPT's response is the same, thus reproducible, for anyone re-running our experiments.
>
> For generalizability (about whether the MoralCoT mechanism can be effective when added to other models): The technical constraint here is that the chain-of-thought design is only compatible with autoregressive LMs that can do text generation. BERT is not a fit here (as it can only do "masked token infilling" for the classification to generate a direct result, but not free text generation for the chain-of-thought subquestion-answering function).
>
> Among all the autoregressive LMs that can do text generation, most of them are not publicly accessible, such as many of Google's large models. Also those models are way too big for us to run in house, and thus we rely on APIs. The key here is that we are testing the limits of what the best/biggest models are capable of; this requires the largest models, which are impossible to run outside of industry without APIs. OpenAI's API is accessible to everyone, and the Bloom model too as of last month. We are taking some time to set them up, and will add the results in a following comment when that's ready, or directly add in our camera ready version.
>
>
>
> > For binary predictions of human moral judgment, how the inter-rater correlation would be? I'm wondering if each model could be comparable to individual's assessment.
>
> Could the reviewer elaborate a bit on the data they are interested in seeing?  Are they interested in 1) the inter-rater reliability (e.g. a metric such as Cohen’s Kappa) for the ~50 subjects that gave judgments on each case?  2) Subject data from each study (for each of the ~150 vignettes, what percent of subjects judged it morally permissible) compared to the model responses for each vignette.  3) Something else.
>
> We are happy to provide either of these measures and particularly agree that 2) could be valuable.  (Another reviewer asked for something similar and we provide an initial set of data in response below.)

---

### Author Response · Authors · 2022-08-02
**Thank you for the overall positive reviews**

We thank all reviewers for their valuable feedback and appreciating our work.

We summarize some of the main strengths of our work as summarized by the reviewers below. The reviewers point out that:
(1) “authors' ideas are quite novel from the psycholinguistic view point” (Reviewer eUQ2),
(2) “Building AI systems that account for human moral judgment is a crucial need”, “The motivation related to AI safety is particularly strong”, “RBQA is a new challenging task/dataset” (Reviewer P8j4), and
(3) “This paper addresses a very significant subject”, “it's a clearly important and original addition”, “The clarity of writing is good” and “The dataset construction is sound and contains a lot of human judgements.” (Reviewer bKwH).

The reviewers have also posed several questions. We have addressed these questions in detail in replies to each comment.

In the revised PDF, we have also incorporated many suggestions about formatting and fixed all the typos. For additional experiments and moving contents from appendix to the main text, we will follow as many suggestions as we can and add them in the one extra page in the camera ready version.

---

### Meta-Review · Area_Chair_taim · 2022-08-30

**Recommendation:** Accept
**Confidence:** Certain

**Metareview:**

This paper addresses an important question of whether LLMs understand human flexible moral judgments. This is a crucial task in AI safety, as it deals with the capability of understanding ethics in relation to heterogeneous contexts. The proposed approach consists in a prompting strategy that generates a sequence of questions and answers that can be exploited to predict human moral judgment.

The reviewers agree that the problem is important and timely, the constructed data resource is sound and of great interest to the community, and the evaluation is done thoroughly. Reviewers' raised concerns and questions are properly addressed by the author's response.

**Award:**

No

---

### Decision · Program_Chairs · 2022-09-14

Accept